# Hematopoietic Stem Cell Transplantation-Associated Neurological Complications and Their Brain MR Imaging Findings in a Pediatric Population

**DOI:** 10.3390/cancers13123090

**Published:** 2021-06-21

**Authors:** Hyewon Shin, Mi-Sun Yum, Min-Jee Kim, Jin Kyung Suh, Ho Joon Im, Hyery Kim, Kyung-Nam Koh, Tae-Sung Ko

**Affiliations:** Asan Medical Center Children’s Hospital, Department of Pediatrics, Ulsan University College of Medicine, Seoul 05505, Korea; hwshin1006@gmail.com (H.S.); pradoxwh@naver.com (M.-J.K.); mint7803@nate.com (J.K.S.); hojim@amc.seoul.kr (H.J.I.); taban@hanmail.net (H.K.); pedkkn@gmail.com (K.-N.K.); tsko@amc.seoul.kr (T.-S.K.)

**Keywords:** pediatrics, hematopoietic stem cell transplantation, neurologic complications, brain MRI

## Abstract

**Simple Summary:**

Neurologic complications following a hematopoietic stem cell transplantation (HSCT) can be caused by various etiologies and significantly contribute to morbidity and mortality. The aim of our retrospective study was to determine the prognostic indicators for HSCT-associated neurological complications in pediatric HSCT recipients using their clinical characteristics and brain magnetic resonance imaging (MRI) lesions. The demographics, received treatments, treatment-related morbidities, laboratory findings and brain MRI findings were reviewed and compared among 51 patients who had underwent a brain MRI due to newly developed neurological symptoms or infection signs during HSCT and follow-up period. Children with neurologic complications associated with infectious causes, malignant disease or severe brain MRI abnormalities were more likely to have poor outcome.

**Abstract:**

Purpose: To determine the prognostic indicators for hematopoietic stem cell transplantation (HSCT)-associated neurological complications, the clinical characteristics and brain magnetic resonance imaging (MRI) lesions in pediatric HSCT recipients were reviewed. Methods: This retrospective study included 51 patients who had underwent a brain MRI due to newly developed neurological symptoms or infection signs during chemotherapy or HSCT. We reviewed the demographics, received treatments, treatment-related morbidities, laboratory findings and brain MRI findings, which were compared between good and poor neurologic outcome groups. Results: Thirty-seven patients (72.5%) fully recovered from the neurologic deficits and fourteen (27.5%) persisted or aggravated. The children with an underlying malignant disease had significantly poorer neurological outcomes (*p* = 0.015). The neurologic complications associated with infection were more frequent in the poor outcome group (*p* = 0.038). In the neuroimaging findings, the extent of the white matter lesions was significantly higher in the poor outcome group, as was that of abnormal enhancement, ventriculomegaly, cortical change, deep gray matter abnormalities and cerebellar abnormalities. Conclusion: Most children with neurologic complications and neuroimaging abnormalities during HSCT had recovered. However, children with neurologic complications associated with infectious causes, malignant disease or severe brain MRI abnormalities should be more carefully monitored during HSCT.

## 1. Introduction

Advances in hematopoietic stem cell transplantation (HSCT), which is widely used to treat malignant and nonmalignant hematologic disorders, genetic disorders, inborn errors of metabolism and autoimmune disorders, have resulted in increased survival rates [1,2,3,4]. Central nervous system (CNS) complications of HSCT are a significant cause of morbidity and significantly contribute to mortality. Approximately 11–59% of patients who undergo HSCT suffer from neurologic complications [5] and more than 90% of the cases who die after HSCT show neuropathological abnormalities in autopsy studies [6]. Various factors, including the pre-transplant conditioning regimen with high-dose chemotherapy, radiation therapy, immunosuppressive therapy, graft versus host disease (GvHD), infection and disease recurrences, are associated with neurotoxicity [7]. Although the neurotoxic mechanisms underlying each cause remain unknown, there are various theories regarding the mechanisms by which chemotherapy and radiotherapy produce damage to the central nervous system which include vascular injury causing chronic ischemia, progressive demyelination of the white matter (WM) and necrosis, oxidative stress and DNA damage, immune dysregulation and stimulation of neurotoxic cytokines [8,9,10].

In addition to the early diagnosis and accurate management of neurologic complications, a prompt assessment of prognostic indicators is also important to reduce the risk of neurologic complications by developing an appropriate treatment plan that will lead to a good outcome. Neuroimaging is a crucial tool in cases with neurologic complications for the early detection of brain lesions, monitoring of disease progression and assessing treatment responses. Additionally, a recent study has found that patients showing structural abnormalities on brain imaging tended to have a higher mortality rate [11]. This result indicates the potential role of neuroimaging as a prognostic modality in patients with neurologic complications.

In our present study, we hypothesized that the various clinical variables and abnormalities that can be detected by neuroimaging would be useful prognostic indicators of the neurologic outcomes in HSCT candidates with neurologic complications. We therefore retrospectively reviewed the clinical data, neuroimaging findings and associated neurologic symptoms among the pediatric patients who underwent HSCT at our hospital and investigated the most optimal prognostic indicators of the neurologic outcomes in these cases. 

## 2. Materials and Methods

### 2.1. Patient Population and Data Collection

From January 2010 to March 2020, 169 children received HSCT and also underwent a brain MRI at the Department of Pediatrics at Asan Medical Center, Seoul, Korea. Among these cases, except 78 patients with previous brain lesion, tumor involvement of CNS and insufficient medical records, 91 patients that had undergone a brain MRI and had newly developed neurological symptoms after the diagnosis of the causative disease were enrolled in this study (Figure 1). We retrospectively reviewed the clinical and laboratory data for these cases and collected the following information: (1) demographics (sex, age at diagnosis, underlying disease, type of transplantation); (2) pre-transplant treatment regimen (systemic and intrathecal chemotherapy, radiation therapy); (3) transplant-associated treatments or complications (immunosuppressant use, GvHD, infection); and (4) the clinical and laboratory characteristics of any neurologic complications.

The chemotherapy, radiotherapy and immunosuppressant regimens which were administered before the onset of neurologic symptoms were investigated. Patients who received a second HSCT were analyzed based on the transplant procedure closest to the onset of neurological symptoms. Electroencephalogram (EEG) results reported by pediatric neurologists were also reviewed. CSF analysis findings were additionally investigated and any abnormal CSF findings were defined as CSF pleocytosis or positive CSF culture and viral PCR and/or the presence of CSF blasts.

### 2.2. Neurological Outcome Assessments

To identify effective prognostic indicators of neurologic outcomes in our current series, the patients were classified into good or poor outcome groups. The patients in the good outcome group showed a disappearance of symptoms with or without improvement in their abnormal findings on brain MRI. Patients with a poor outcome was defined as cases with the persistent neurologic sequelae including a bedridden state or death.

### 2.3. Classification and Scoring of the Neuroimaging Data 

The brain MRI findings for our study cohort reported by pediatric neuroradiologists were reviewed. An abnormal lesion was classified as frontal, temporal, corpus callosum, inferomedial, or parieto-occipital and further separated into 14 subdivisions according to the location of the white matter (WM) injury. The number of these WM subdivisions with T2 fluid attenuated inversion recovery (FLAIR) high signal intensities was counted. In addition to the extent of the WM lesions, unidentified scattered spots, abnormal enhancements, ventriculomegaly, cortical changes, deep gray matter abnormalities and cerebellar abnormalities in the T2 FLAIR, or gadolinium enhancement, were regarded as additional findings and investigated. (Table 1) The extent of the WM lesions and existence of other MRI findings were compared between the two neurologic outcome groups.

### 2.4. Statistical Analysis

SPSS version 24 (SPSS, Chicago, IL, USA) was used for all statistical analysis of the data. A chi-square test and Fisher’s exact test were used for the univariate analysis of prognostic factors of neurological symptoms, with continuous variables compared using the Student’s *t* test. Logistic regression analysis was used to evaluate the association between patient outcomes and neuroimaging findings. *p*- values of less than 0.05 were considered to indicate statistical significance.

## 3. Results

Among the 91 pediatric patients in our present cohort who underwent neuroimaging testing due to clinical symptoms, 51 patients (51/91, 56.0%) showed abnormal findings. The baseline characteristics, treatments and clinical and laboratory findings of neurological complications for the 51 study patients are listed in Table 2.

### 3.1. Baseline Demographics and Treatments of the Study Cohort

Twenty-seven of the children in our study series were male (52.9%) and the mean age at diagnosis of the underlying disease was 7.7 years. The underlying diseases in this pediatric cohort included malignancy in 39 patients (76.5%) and non-malignant disease in 12 cases (23.5%). In terms of the transplantation types, 23 children (45.1%) received haplotype HSCT, 20 (39.2%) received allogenic HSCT from an unrelated donor and 8 (15.7%) received allogenic HSCT from a related donor. Abnormal neuroimaging findings were evident in 19 patients (37.3%) prior to HSCT and in the remaining 32 cases (62.7%) after HSCT. 

Cyclophosphamide (66.7%), cytarabine (64.7%), etoposide (52.9%) and vincristine (51.0%) were the commonly administered neurotoxic chemotherapeutics among the children with abnormal imaging findings and 36 of these children (70.6%) received intrathecal chemotherapy with cytarabine, methotrexate and hydrocortisone. Cranial irradiation was done in 22 patients (43.1%) prior to symptom onset and almost 50% of the children received immunosuppressants including cyclosporine, tacrolimus (FK506), or mycophenolate mofetil (MMF).

### 3.2. Clinical and Laboratory Characteristics Associated with Neurologic Complications

The mean age of onset of the neurological or infectious symptoms and signs in our 51 pediatric cases with abnormal findings was 10.2 years. Among the 32 patients with abnormal brain MRI findings after HSCT, the median duration from transplantation to symptom onset was 178.5 days, but this onset occurred within 100 days after HSCT in almost 50% of the patients. Motor symptoms most commonly presented, arising in 20 patients (39.2%), followed by decreased consciousness (12 patients; 23.5%) and headache (10 patients;19.6%). Five patients (9.8%) underwent a brain MRI due to infection signs without neurological symptoms.

An electroencephalogram (EEG) was performed in 21 children, of which 4 (4/21, 19%) had normal EEG findings, 17 (17/21, 81%) had EEG abnormalities; 6 had focal epileptiform discharge and 11 had cerebral dysfunction. There were 28 patients who underwent a CSF test, of which 14 (14/28, 50%) showed normal CSF findings, 12 (12/28, 42.9%) had CSF abnormalities and 2 (2/28, 7.1%) had newly identified CSF blasts. Details of CSF abnormalities are described in Appendix C. Treatment associated neurological complications were the most common cause of neurologic complications (33 patients; 64.7%), followed by infection (14 patients; 27.5%) and newly involved malignancy (4 patients; 7.8%) (Figure 2).

### 3.3. Neurologic Outcome Assessments

We compared the clinical and laboratory findings between the 37 children (72.5%) with a good outcome and 14 (27.5%) with a poor outcome (Table 3), among the patients showing abnormal findings. Baseline demographics including sex, median age at diagnosis and type of transplantation, were not significantly different between these two groups. On the other hand, a malignancy was significantly associated with a poor outcome (25/37, 67.5% vs. 14/14, 100%; *p* = 0.015). In terms of the factors associated with the treatments received by the children, there was no statistically significant association found between systemic and intrathecal chemotherapy and outcomes. Cranial radiation was also not significantly associated with a poor outcome (13/37, 35.1% vs. 9/14 64.2%; *p* = 0.061).

Among the initial presenting neurological symptoms in our cohort, the children with a decreased consciousness were more likely to have a poor outcome, with statistical significance (6/37, 16.2% vs. 6/14, 42.9%; *p* = 0.045). There was no difference in the EEG and CSF abnormalities between the good and poor outcome groups. Infection was found to be a more frequent cause of neurological symptoms in the poor outcome group (7/37, 18.9% vs. 7/14, 50%; *p* = 0.038). The causative microorganism was identified in eight of 14 patients whose neurological symptoms and brain MRI abnormalities were due to an infection, i.e., 3 bacterial infections (*L. monocytogenes*, one case; S. viridians group, two cases), five viral infections (CMV, three cases; adenovirus, one case; BK virus, one case) and an unidentified pathogen in six patients with clinical CNS infections, including three with a brain abscess evident by MRI.

Among the 32 patients that showed neurological symptoms after HSCT, the HSCT-associated factors were compared between the 21 children in this subpopulation with a good neurologic outcome and the 11 remaining cases with a poor neurologic outcome (Table 4). Neither the GvHD nor the type of immunosuppressant affected the neurologic prognosis. More of the children in the poor outcome group had a neurological symptom onset later than 100 days after HSCT.

### 3.4. Comparison of MRI Findings

The extent of the WM lesions was more prevalent in the poor outcome group. Among the other accompanying abnormal brain MRI findings, abnormal enhancements, ventriculomegaly, cortical changes, deep gray matter abnormalities and cerebellar abnormalities were more frequent in the poor outcome group, with statistical significance (Table 5).

There was no statistically significant difference in outcome between the patients with normal MRI and abnormal MRI findings (Appendix A). However, in patients with normal MRI findings, persistent neurological symptoms such as focal seizure and headache were relatively mild and there was no death due to neurological complications (Appendix B).

In our current study series, six children were diagnosed with PRES, but five of these cases had a good prognosis. All six children diagnosed with PRES had hypertension and neurological symptoms (four with seizures, two with a decreased consciousness) and MRI findings of frontoparietal or parieto-occipital lesions. The extent of the WM lesions according to the classification and scoring in the PRES cases in our current series was 1–2 in the five recovered patients and 5 in the remaining female patient with a poor outcome and that had a tiny petechial hemorrhage in the involved region. This patient had no coagulopathy but mild thrombocytopenia, which was also seen in one of the five recovered patients. She eventually died of multiorgan failure following GI bleeding and pancreatitis caused by uncontrolled GvHD.

## 4. Discussion

This study retrospectively investigated children that had received HSCT and subsequently underwent a brain MRI due to newly developed neurological symptoms or infection signs during chemotherapy or HSCT. Among the patients in our pediatric cohort, 56% showed abnormal brain MRI findings and we attempted to identify associated factors in accordance with the neurological outcomes of these cases. In our poor outcome group, there were more instances of malignancy as the underlying disease and infection as the cause of the neurological symptoms. Notably, however, there was no differences found in the factors associated with transplantation (immunosuppressants and GvHD) according to the outcome. On the other hand, the severity of the abnormal brain MRI findings was significantly associated with the clinical outcomes in our cohort.

Previous studies [2,12,13,14,15,16] have reported that about 8.3–24% of patients developed neurological symptoms after HSCT. These neurological complications, including encephalopathy, have been associated with a poor prognosis resulting in a high mortality rate in children after HSCT [6,17] and an accurate and timely diagnosis and intervention is crucial to improve the outcomes in these cases. Many of the currently used anticancer drugs and myeloablative drugs administered for HSCT have neurologic side effects [18]. Although it is difficult to find a specific causal relationship between a hazardous dosage or a particular chemotherapeutic agent and CNS toxicity, the mechanisms of neurotoxicity caused by chemotherapy are suggested by several hypotheses [19]. In our present pediatric cohort, an underlying malignancy was associated with poor neurological outcomes, among various clinical factors. Although there were no significant associations identified by our present analyses due to small number of patients examined, our results indicated that a higher dose of chemotherapy, TBI doses and immunosuppressants in patients with malignant disease can be associated with a poor neurologic prognosis, consistent with previous studies [6,7,12,13,14,15,16,20].

Leukoencephalopathy is one of the major side effects of chemotherapy but can be mild and reversible in many cases [19]. Posterior reversible encephalopathy syndrome (PRES), caused by an acute rise in blood pressure associated with calcineurin inhibitors and corticosteroids, is a clinical syndrome typically characterized by vision changes, altered mental status and seizures. Although the prognosis of PRES is known to be favorable in most cases, neurological sequelae can persist in 10–20% of affected patients [21]. Many studies have attempted to identify the etiology and prognostic factors and associated brain MRI findings, for PRES, but there are still no established criteria for this condition. The risk factors identified by each prior study of this disorder have been different, but the following variables have been suggested to be associated with poor outcomes: severe encephalopathy, hypertensive etiology, hyperglycemia, neoplastic etiology, longer time to control the causative factor, presence of multiple comorbidities, elevated CRP, low CSF glucose, coagulopathy and brain MRI findings of corpus callosum involvement, extensive cerebral edema or worsening imaging severity, hemorrhage, subarachnoid hemorrhage and restrictive diffusion [22].

Infection was found in our current analysis to be a more frequent cause of neurologic complications in the poor outcome group. Children who have undergone HSCT are particularly susceptible to viruses, bacteria, or invasive molds and CNS infection has been shown to be indicative of a poor prognosis [2,15,16,20,23]. The incidence of CNS infection after allo-HSCT can be as high as 15% [24]. Aspergillus and toxoplasma are one of the most common causes and other fungi and viruses can also be a source CNS infection. Prior studies have indicated that a high-risk disease status is a risk factor for CNS infection, which in turn significantly reduces the overall survival rate [2,15]. Hence, if any indicators of CNS infection or neurological symptoms are present, a rapid diagnostic test and treatment including antimicrobial agents should be conducted with careful monitoring and concerns. However, there are several limitations to identifying the causative microorganisms in practice because a brain biopsy can be problematic for patients with a poor underlying condition and PCR detection is only possible for a few CSF viruses, such as HSV and CMV, despite the high sensitivity and specificity of this test [25]. Hence, empirical treatments should be initiated depending on the time since HSCT and the extent of the host immune deficiency to improve patient outcomes.

In many previous studies, GvHD has been identified as a risk factor for neurologic sequelae [12,13,25,26,27] and the prognosis of CNS GvHD is alleged to be very poor [28]. In addition to CNS GvHD, CNS infections and thrombotic microangiopathy (TMA)-associated neurological events can be frequent following the increased use of calcineurin inhibitors and corticosteroids in patients with GvHD [29]. In our current study, none of the children were diagnosed with CNS GvHD and the presence of GvHD or the use of immunosuppressant showed no significant association with poor clinical outcomes in our cohort. As CNS GvHD can be diagnosed only when drug toxicity or infection are ruled out, or when the symptoms meet other facultative criteria including abnormal MRI, or cerebrospinal fluid (CSF) abnormalities and pathology revealing GvHD lesions [29], the reported incidence of CNS GvHD can be very rare. Moreover, the differential diagnosis of causality in patients with GvHD and neurological symptoms is very difficult and the risk of immunosuppression and GvHD are always intermixed. This ambiguous causality can result in no significant correlation being found between these factors and clinical outcomes. In addition, it should be noted that our present study series included relatively small number of children with neurological symptoms after HSCT.

Many previous studies have attempted to interpret the changes in brain MRI findings after chemotherapy or HSCT. In a prior systematic review of chemotherapy-induced changes in the brain and cognitive functioning by Mingmei Li et al., neuroimaging data indicated a reduced grey matter density in cancer patients in the frontal, parietal and temporal regions. Data from diffusion-weighted MRI in that report suggested reduced WM integrity involving the superior longitudinal fasciculus, corpus callosum, forceps major and corona radiata and altered structural connectivity across the whole brain network and moderate-to-strong correlations between worsening cognitive function and morphological changes in the frontal brain regions [30]. There have been reports that diffusion tensor imaging (DTI) is a more useful tool than conventional MRI to confirm changes in WM integrity in patients undergoing HSCT [31,32,33,34]. In addition, in previous studies using voxel-based morphometry (VBM), a reduced volume of white and gray matter was observed in patients who underwent chemotherapy [35,36]. However, those investigations did not deal with clinically significant neurological events during and after HSCT, but preclinical quantitative imaging data and cognitive function. To support the clinical diagnosis and treatment of neurological complications of HSCT, we here compared brain MRI findings according to clinical outcomes. In the brain MRI findings, we identified a statistically significant relationship between a poor outcome and a more severe WM injury, abnormal enhancement, ventriculomegaly, cortical change, deep gray matter abnormality, or cerebellar abnormality. On the other hand, only unspecified scattered spots on the MRI were associated with a good outcome in our current pediatric cohort. Hence, along with the various prognostic factors discussed above, a brain MRI is an important prognostic tool in these cases. Moreover, these MRI findings can be combined with other indicators to develop a scoring system for predicting the prognosis.

There were several limitations of our current study of note. There was a selection bias for patients who had undergone a brain MRI. It is difficult to accurately describe the incidence of neurological complications because patients with neurological complications were not first selected. However, in our center, when clinically significant neurological symptoms occur, brain MRI is performed on most patients, so except for children with minor and transient neurological symptoms, most would have been included. The diagnosis of neurological symptoms in these cases was based on clinical and imaging findings. A definitive etiology could therefore not be identified. The cumulative dose of chemotherapeutics, radiation and immunosuppressants was not investigated although the prognosis would have been affected by these parameters. Due to the retrospective design of this study, many patients who were transferred to our center due to refractory diseases were excluded because there were no available data on their previous treatments. In addition, minor neurocognitive dysfunction in patients in our good outcome group, or patients who had no evident abnormalities on a conventional brain MRI scan, could not be evaluated. A further, well-designed, prospective large multicenter study will be needed. 

## 5. Conclusions

Neurological complications and associated brain MRI abnormalities including WM injury are frequent events in children following HSCT and treatments for cancer, but most show a complete recovery. However, children with underlying malignant disease or with neurologic symptoms due to infectious causes should be carefully monitored. In addition, the brain MRI findings for these cases, particularly the extent of WM injury, are associated with their clinical outcomes. A model that predicts the prognosis of children after HSCT in accordance with their brain MRI findings would be very useful. Future large-scale multicenter studies and subgroup analyses of the prognostic indicators of neurological symptoms after HSCT are warranted.

## Figures and Tables

**Figure 1 cancers-13-03090-f001:**
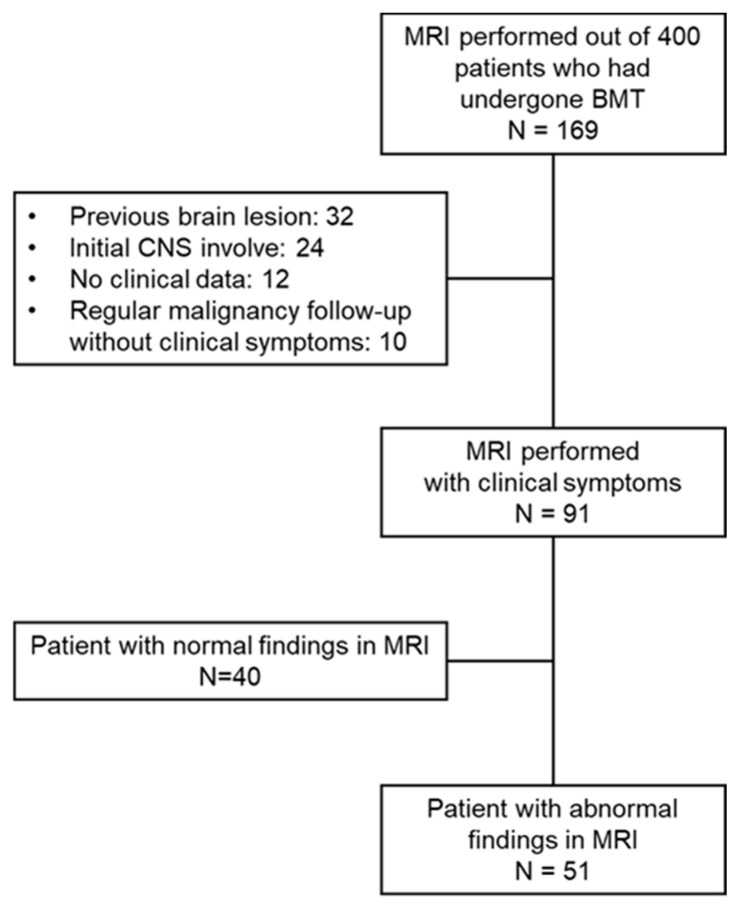
Study scheme.

**Figure 2 cancers-13-03090-f002:**
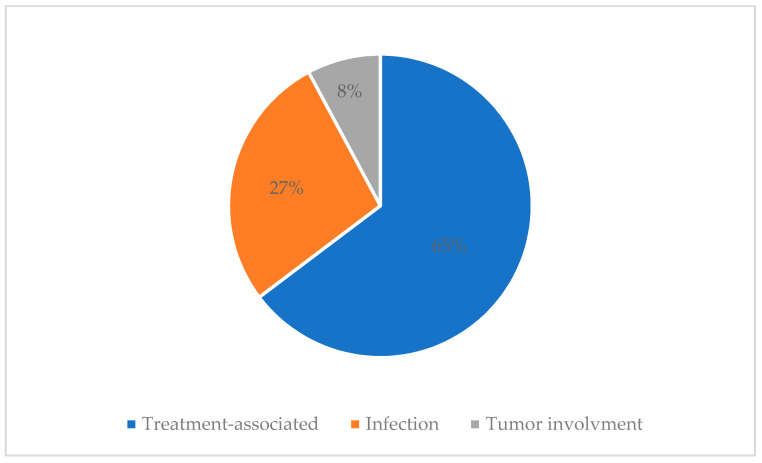
Causes of neurological complications.

**Table 1 cancers-13-03090-t001:** Division of white matter lesions on brain MRI findings.

Location of WM Lesions	Subdivision
Frontal	Subcortical
	Periventricular
	Central
Temporal	Subcortical
	Periventricular
	Central
Corpus callosum	Genu
	Body
	Splenium
Inferomedial	Internal capsule
	Brainstem
Parieto-occipital	Subcortical
	Periventricular
	Central
**Other findings**	
Unidentified scattered spots	
Abnormal enhancement	
Ventriculomegaly	
Cortical change	
Deep gray matter abnormalities	
Cerebellar abnormalities	

**Table 2 cancers-13-03090-t002:** Baseline characteristics of the study cohort and received treatments.

Total	51
Sex, *n* (%)	
Male	27 (52.9%)
Female	24 (47.1%)
Median age at diagnosis, year	
Mean ± SD (range)	7.6 ± 6.0 (0–17)
Underlying disease, *n* (%)	
Malignant disease	39 (76.5%)
ALL (acute lymphoblastic leukemia)	19 (37.3%)
AML (acute myeloid leukemia)	12 (23.5%)
MPAL (mixed phenotype acute leukemia)	2 (3.9%)
CML (chronic myelogenous leukemia)	1 (2.0%)
JMML (juvenile myelomonocytic leukemia)	1 (2.0%)
MDS (myelodysplastic syndrome)	3 (5.9%)
Rhabdomyosarcoma	1 (2.0%)
Non-malignant disease	12 (23.5%)
SAA (severe aplastic anemia)	4 (7.8%)
HLH (Hemophagocytic lymphohistiocytosis)	4 (7.8%)
CDA (congenital dyserythropoietic anemia)	2 (3.9%)
Hypermegakaryocytic thrombocytopenia	1 (2.0%)
XLP1 syndrome (X-linked lymphoproliferative syndrome type 1)	1 (2.0%)
Type of transplantation, *n* (%)	
HaploPBSCT	23 (45.1%)
HLA-matched unrelated donor alloPBSCT	17 (33.3%)
Single allele HLA-mismatched unrelated donor alloPBSCT	3 (5.9%)
HLA-matched related donor alloPBSCT	8 (15.7%)
Time of onset of neurologic complications, *n* (%)	
Before HSCT	19 (37.3%)
After HSCT	32 (62.7%)
Systemic chemotherapy, *n* (%)	
Cyclophosphamide	34 (66.7%)
Cytarabine	33 (64.7%)
Etoposide	27 (52.9%)
Vincristine	26 (51.0%)
Methotrexate	16 (31.4%)
Ifosfamide	12 (23.5%)
Blinatumomab	3 (5.9%)
Cisplatin	1 (2.0%)
Carboplatin	1 (2.0%)
Vinblastine	1 (2.0%)
Intrathecal chemotherapy, *n* (%)	
Not done	15 (29.4%)
Done	36 (70.6%)
Cytarabine	33 (64.7%)
Methotrexate	26 (51.0%)
Hydrocortisone	15 (29.4%)
Cranial radiation, *n* (%)	22 (43.1%)
Immunosuppressant administration, *n* (%)	
Cyclosporine	19 (37.3%)
FK506	13 (25.5%)
MMF	13 (25.5%)
None	25 (49.0%)
Onset age of symptoms, year	
Mean ± SD (range)	10.2 ± 6.7 (0–24)
Duration from BMT to symptoms, day	
Mean ± SD (range)	399.8 ± 593 (22–2942)
Median	178.5
≤100day, *n* (%)	10 (19.6%)
>100day, *n* (%)	22 (43.1%)
Initial presenting symptoms, *n* (%)	
Motor symptom	20 (39.2%)
(seizure, ataxia, gait disturbance, tremor, weakness, spasms)	
Decreased consciousness	12 (23.5%)
Headache	10 (19.6%)
Brainstem dysfunction (ptosis, dysarthria)	2 (3.9%)
Visual symptom	2 (3.9%)
Infection symptom	5 (9.8%)
EEG finding, Total, *n* (%)	21
Normal	4 (19.0%)
Abnormal	17 (81.0%)
Focal epileptiform discharges	6 (28.6%)
Cerebral dysfunctions	11 (52.4%)
CSF analysis, Total, *n* (%)	28
Normal	9 (32.1%)
Abnormal	17 (60.7%)
CSF blast	2 (7.1%)
Cause of neurologic complications, *n* (%)	
Treatment associated	33 (64.7%)
Infection	14 (27.5%)
Tumor involvement	4 (7.8%)

**Table 3 cancers-13-03090-t003:** Neurological outcome assessments in patients with MRI abnormalities.

Total, *n* (%)	Good Outcome, *n* = 37 (72.5%)	Poor Outcome, *n* = 14 (27.5%)	*p-Value*
Sex, *n*			0.129
Male	22	5	
Female	15	9	
Mean age at diagnosis, year			0.975
Median [IQR]	6	8.5	
	[2.00, 14.00]	[1.25, 13.75]	
Underlying disease, *n*			*0.015*
Malignant	25	14	
Non-malignant	12	0	
Type of transplantation, *n*			0.946
HaploPBSCT	6	2	
Unrelated donor alloPBSCT	14	6	
Related donor alloPBSCT	17	6	
Time at appeared neurologic complications, *n*			0.15
Before HSCT	16	3	
After HSCT	21	11	
Systemic chemotherapy, *n* (%)			
Cyclophosphamide	25	9	0.824
Cytarabine	21	12	0.053
Etoposide	17	10	0.104
Vincristine	20	6	0.475
Methotrexate	13	3	0.346
Ifosfamide	8	4	0.602
Blinatumomab	3	0	0.272
Cisplatin	1	0	1
Carboplatin	1	0	1
Vinblastine	1	0	1
Intrathecal chemotherapy, *n* (%)			0.145
Not done	14	2	
Done	24	12	
Cytarabine	22	11	0.202
Methotrexate	19	7	0.931
Hydrocortisone	11	4	0.935
Cranial radiation, *n* (%)	13	9	0.061
Initial presenting symptoms, *n* (%)			
Motor symptoms	14	6	0.743
(seizure, ataxia, gait disturbance, tremor, weakness, spasm)			
Decreased consciousness	6	6	*0.045*
Headache	8	2	0.556
Brainstem dysfunction (ptosis, dysarthria)	2	0	1
Visual symptoms	2	0	1
Infection symptoms	5	0	0.305
Cause of neurologic complications, *n* (%)			0.082
Treatment-associated	27	6	0.057
Infection	7	7	*0.038*
Tumor involvement	3	1	1
CSF analysis, Total, *n*	19	9	0.826
Normal	10/19	4/9	
Abnormal	8/19	4/9	
CSF blast	1/19	1/9	

**Table 4 cancers-13-03090-t004:** Neurological outcome assessments in patients with neurological symptom onset after HSCT.

Total, *n* (%)	Good Outcome, *n* = 21 (65.6%)	Poor Outcome, *n* = 11 (34.4%)	*p*-Value
Conditioning intensity, *n*			0.703
Reduced intensity conditioning	15	7	
Myeloablative conditioning	6	4	
Total body irradiation (TBI) based conditioning, *n*	10	6	1
Immunosuppressant administration, *n*			
Cyclosporine	12	7	0.722
FK506	7	6	0.246
MMF	8	5	0.687
None	5	2	0.715
GVHD, *n*	14	5	0.246
Onset of symptoms after HSCT, day			
Median [IQR]	120	234	0.275
	[78.00, 485.00]	[160.50, 486.00]	
≤100 day, *n*	9	1	*0.05*
>100 day, *n*	12	10	

**Table 5 cancers-13-03090-t005:** MRI findings according to neurologic outcome.

	Good Outcome, *n* = 37 (72.5%)	Poor Outcome, *n* = 14 (27.5%)	*p*-Value	OR (95% CI)	*p*-Value
Extent of WM lesions			*<0.001*	2.71 (1.47~5.01)	*0.001*
1	23	1			
2	7	4			
3	6	1			
4	1	0			
5	0	1			
6	0	3			
7	0	2			
9	0	2			
Unidentified scattered spots	23	1	*<0.001*	0.05 (0.01~0.4)	*0.005*
Abnormal enhancement	9	8	*0.027*	4.15 (1.13~15.19)	*0.032*
Ventriculomegaly	2	8	*<0.001*	23.33 (3.95~137.64)	*<0.001*
Cortical change	14	12	*0.002*	9.86 (1.92~50.7)	*0.006*
Deep gray matter abnormalities	2	6	*0.001*	13.12 (2.22~77.42)	*0.004*
Cerebellar abnormalities	2	5	*0.005*	9.72 (1.61~58.57)	*0.013*

## Data Availability

The data presented in this study are available on request from the corresponding author.

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
