# Peer review of "Hematopoietic Stem Cell Transplantation-Associated Neurological Complications and Their Brain MR Imaging Findings in a Pediatric Population"

_cancers, 2021, doi:10.3390/cancers13123090_

Round 1

Reviewer 1 Report

Shin et al report a retrospective single-center analysis of 91 children with newly developed neurologic complications during or before allogeneic stem cell transplantation (HSCT) for malignant or non-malignant diseases with available data on brain MR imaging.

The aim of this study was to identify neuroimaging abnormalities as reliable predictors of post-transplant neurologic outcome in a subgroup of 51 children, who had confirmed brain MRI abnormalities. In addition to various clinical transplant characteristics, electroencephalographic findings and cerebrospinal fluid analyses were evaluated.

According to the neurologic outcome, the investigators separated a good risk group with complete resolution of neurologic symptoms (n=37) from a poor risk group of patients (n=14), who faired inferior or even succumbed. Comparing the MRI findings of both groups, they found significantly more white-matter lesions and other brain abnormalities in patients with poor neurologic outcome. Furthermore, they found that an altered level of consciousness, detection of infections or HSCT for malignancies had a significant impact on neurologic outcome. Administration of cranial irradiation, intrathecal chemotherapy, the presence of EEG abnormalities or abnormal findings of the CNS fluid did not seem to affect the neurologic outcome in these patients.

Immunosuppressive drugs (tacrolimus, cyclosporine A, MMF) or graft versus host disease (GvHD) did not affect neurologic outcome in a further subgroup of 32 patients who developed their neurologic symptoms after HSCT.

Albeit providing clinically relevant information, the manuscript has considerable limitations:

1 There is a strong selection bias in this cohort for only patients with neurologic symptoms and available brain MRI were included. How many of the 400 patients who received their HSCT between 2010 and 2020 had pre-existing neurologic complications but did not undergo brain imaging? How many patients developed their neurologic complications after the conditioning for HSCT?

2 The authors should provide clinical and outcome data of the 40 patients (as shown in figure 1) with neurologic symptoms who had normal MRI findings. If the MRI alterations described in the present manuscript are reliable prognostic parameters, this group should have fared significantly better as compared to the group of patients with MRI abnormalities. 

3 The authors decided to classify the brain MRI abnormalities primarily according to the number and location of white matter lesions and additional findings such as ventricular enlargement, cortical changes or cerebellar abnormalities. They should explain why they did not consider including other common radiologic criteria such as large- or small vessel abnormalities, periventricular enhancement, edema or cerebral atrophy.

4 Given that infections seem to play a major role in the poor prognosis group, the microbiologic testing results and CSF analysis data should be provided for both patient groups in table 3.

5 May I suggest including the conditioning intensity (reduced/RIC versus myeloablative conditioning/MAC) as additional variable in table 3. Did the authors also use a total body irradiation (TBI-) based conditioning (e.g. in children with ALL)? If yes, they should further separate between TBI-based versus chemotherapy based conditioning.

6 The discussion should be shortened for a better reading of the manuscript. For example, an extended section on posterior reversible encephalopathy (PRES) is provided (line 217-240), where the authors have included a clinical description of their six affected children. This part can be moved to the results section.

The same is true for the part on infections (line 241-261). I suggest also to move this to the results section and to include an additional table on detected microorganisms.

Minor:

7 The degree of HLA matching should be included in the donor section, e.g. HLA-matched unrelated donors (10/10).

8 There is a typing error in Figure 2.

Reviewer 2 Report

This study describes a cohort of pediatric patients undergoing allogeneic stem cell transplantation who underwent a brain MRI: what where the patient characteristics, findings on MR and what was the prognosis of these patients. 

The basic limitation of this study is: what can other clinicians learn from these findings ? Looking at the case mix of the population of children undergoing HSCT in this centre (table 2) this appears a representative SCT centre (400 allow in 10 years). 169 MRI scans seems a fairly high number of scans in this population. Does this imply that scans are offered quite frequently in this centre ? 91 of 169 scanned patients had symptoms. Decreased  mentality is mentioned (3.2) as a symptom. Is this decreased consciousness ? Please rephrase. The authors findings indicate that a combination of underlying disease (malignancies), clinical findings (infections) in combination with white matter injury on MRI lead to a poor prognosis. defined as persistent neurological sequelae, including bedridden state and death. 

Round 2

Reviewer 1 Report

The authors have addressed the raised points accordingly and provided additional data as suggested.